# Rapid population decline in migratory shorebirds relying on Yellow Sea tidal mudflats as stopover sites

Colin E. Studds[1,2,3], Bruce E. Kendall[4], Nicholas J. Murray[1,5], Howard B. Wilson[1], Danny I. Rogers[6], Robert S. Clemens[1], Ken Gosbell[7], Chris J. Hassell[8], Rosalind Jessop[9], David S. Melville[10], David A. Milton[11], Clive D.T. Minton[7], Hugh P. Possingham[1,12], Adrian C. Riegen[13], Phil Straw[14], Eric J. Woehler[15] & Richard A. Fuller[1]

Migratory animals are threatened by human-induced global change. However, little is known about how stopover habitat, essential for refuelling during migration, affects the population dynamics of migratory species. Using 20 years of continent-wide citizen science data, we assess population trends of ten shorebird taxa that refuel on Yellow Sea tidal mudflats, a threatened ecosystem that has shrunk by >65% in recent decades. Seven of the taxa declined at rates of up to 8% per year. Taxa with the greatest reliance on the Yellow Sea as a stopover site showed the greatest declines, whereas those that stop primarily in other regions had slowly declining or stable populations. Decline rate was unaffected by shared evolutionary history among taxa and was not predicted by migration distance, breeding range size, non-breeding location, generation time or body size. These results suggest that changes in stopover habitat can severely limit migratory populations.

[1] School of Biological Sciences, University of Queensland, St. Lucia, 4072 Queensland, Australia. [2] Department of Geography and Environmental Systems, University of Maryland, Baltimore County, Baltimore, Maryland 21250, USA. [3] Migratory Bird Centre, Smithsonian Conservation Biology Institute, Washington, District Of Columbia 20008, USA. [4] Bren School of Environmental Science & Management, University of California, Santa Barbara, California 93106, USA. [5] Centre for Ecosystem Science, School of Biological, Earth and Environmental Sciences, University of New South Wales, Sydney, 2052 New South Wales, Australia. [6] Arthur Rylah Institute for Environmental Research, Heidelberg, 3034 Victoria, Australia. [7] Victorian Wader Study Group, 165 Dalgetty Rd., Beaumaris, 3193 Victoria, Australia. [8] Global Flyway Network, PO Box 3089, Broome, 6725 Western Australia, Australia. [9] Phillip Island Nature Park, PO Box 97 Cowes, 3922 Victoria, Australia. [10] Ornithological Society of New Zealand, 1261 Dovedale Road, RD 2 Wakefield, Nelson 7096, New Zealand. [11] Queensland Wader Study Group, c/o CSIRO Oceans and Atmosphere, PO Box 2583, Brisbane, 4001 Queensland, Australia. [12] Department of Life Sciences, Imperial College London, Silwood Park, Ascot SL5 7PY, UK. [13] Ornithological Society of New Zealand, 231 Forest Hill Road, Waiatarua, Auckland 0612, New Zealand. [14] Avifauna Research and Services Pty Ltd, PO Box 2006, Rockdale, 2216 New South Wales, Australia. [15] Institute for Marine and Antarctic Studies, University of Tasmania, Sandy Bay, 7001 Tasmania, Australia. Correspondence and requests for materials should be addressed to C.E.S. (email: studdsc@umbc.edu).

nimal migrations are among the greatest spectacles of the natural world, yet they are increasingly imperilled by human activity[1]. Animals that undergo long-distance migrations spend most of their time in widely separated breeding and non-breeding habitats. As a result, research on population abundance has focused on these two phases of the annual life cycle, where demographic impacts[2–4] and especially habitat loss[5–7] are considered the leading cause of population declines. Migratory animals must undertake energetically demanding migrations covering thousands of kilometres between breeding and non-breeding sites, and many interrupt their journeys to rest and refuel at stopover sites along the way. Such sites have a potential importance disproportionate to their area and the amount of time they are used, and quantitative models indicate that the quantity and quality of stopover habitat can limit population abundance[8–10]. However, there are few empirical studies linking population abundance to processes, such as habitat loss, experienced during migration[11–13].

There is growing concern that loss and degradation of stopover habitat is causing population declines of migratory shorebirds of the East Asian-Australasian Flyway (EAAF). Millions of shorebirds migrate annually through the EAAF from breeding grounds in China and the Russian and Alaskan Arctic to non-breeding habitats in Southeast Asia, Australia and New Zealand. The birds congregate at high densities on tidal mudflats in the Yellow Sea region of East Asia, using them as staging sites[14] to refuel as they complete tight annual schedules of migration, reproduction and moult[15]. However, the Yellow Sea is increasingly being degraded and >1% of its tidal mudflats are destroyed annually for coastal development[16], leading to widespread concern that collapse of this ecosystem is driving migratory species' declines[17]. Indeed, shorebirds in the EAAF are undergoing severe, long-term reductions in abundance on stopover and non-breeding sites[11,18–21]. Although the Yellow Sea region is a migration bottleneck for many EAAF shorebirds, species vary in their reliance on migratory stopover sites in this area. At one extreme, 100% of the *menzbieri* subspecies of bar-tailed godwit (*Limosa lapponica*) refuel at stopover sites in the Yellow Sea. Conversely, grey-tailed tattler (*Tringa brevipes*) refuels at a wide range of sites, with only 3% of the population staging in the Yellow Sea[22–24]. This interspecific variation in the use of Yellow Sea tidal mudflats

provides a unique opportunity to investigate whether deteriorating and disappearing stopover habitat can drive population declines.

Here, we measure abundance and population trends for ten EAAF shorebird taxa for which expert-defined migratory connectivity networks were available[22] to test if taxa that rely heavily on Yellow Sea tidal mudflats to stage their long-distance migrations are experiencing the most rapid population declines. The migratory connectivity networks comprise nodes that identify spatially distinct populations of each taxon and edges that trace their unique migration routes through the EAAF. We also evaluate if conditions in the non-breeding grounds contribute to population trends. Our data were non-breeding season counts made between 1993 and 2012 across Australia and New Zealand at 43 sites organized into four nodes of the migratory connectivity network (Supplementary Table 1). We calculated reliance on the Yellow Sea by tallying the proportion of each taxon's population that refuels in Yellow Sea nodes of the migratory connectivity network. Finally, we evaluated the importance of four common predictors of population decline and extinction risk[25,26]: migration distance, breeding range size, generation time and body size.

If threats in the Yellow Sea are driving population trends, taxon-specific changes in abundance should be correlated with Yellow Sea reliance during migration, and taxon-specific trends should vary little among non-breeding nodes. Alternatively, if environmental conditions in non-breeding areas affect population trends, taxon-specific changes in abundance should vary markedly among non-breeding nodes, and taxon-specific trends should be only weakly related to Yellow Sea reliance during migration. Under this latter scenario, taxon-specific declines might be expected to vary according to local differences in habitat change, rather than show consistency among nodes.

## Results

**Predicting variation in flyway-level population trends**. After accounting for the shared evolutionary history among taxa, we found that Yellow Sea reliance was the single most important predictor of variation in population trends (Fig. 1a). Population trends were strongly negatively related with Yellow Sea reliance

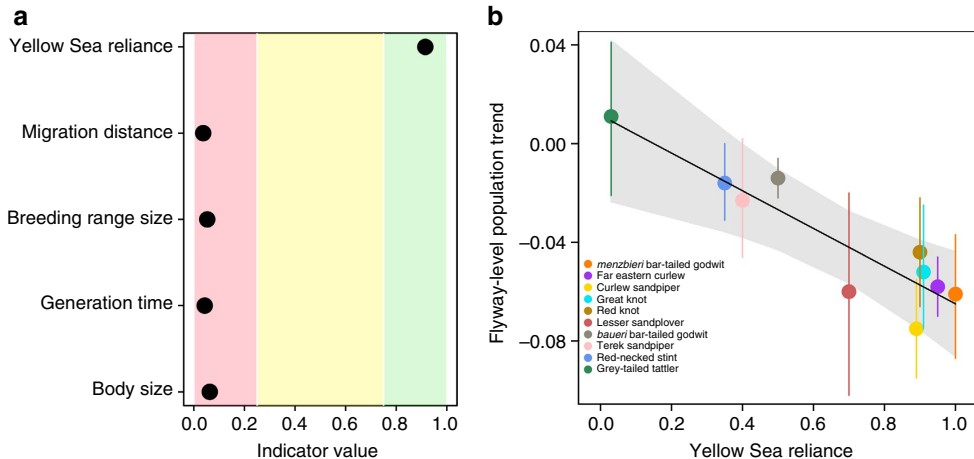

**Figure 1 | Predictors of flyway-level population trend estimates between 1993–2012 for ten EAAF migratory shorebirds.** (**a**) Bayesian variable selection identifying predictors that are important (indicator value ≥ 0.75; green shading), inconclusive (indicator value between 0.25 and 0.75; yellow shading) and unimportant (indicator value ≤ 0.25; red shading). (**b**) Bayesian linear regression of Yellow Sea reliance as a predictor of flyway-level population trend estimates. Grey shading shows the 95% CRI around the regression line. Points show flyway-level population trend estimates, the mean annual rate of change in total abundance estimates. Error bars represent the 95% CRI around population trends. These analyses included the *menzbieri* subspecies of bar-tailed godwit and excluded the *baueri* subspecies (see Methods).

(slope $= -0.92$, 95% credible interval (CRI) $= -1.25$, $-0.59$; Fig. 1b), indicating that long-term population declines were greatest for taxa that depend more heavily on refuelling stops in this region (Table 1; Fig. 2). Taxa with $\leq 40\%$ of the population relying of the Yellow Sea remained stable or declined only slightly, with an average change of $-1.0\%$ year$^{-1}$

(95% CRI $= -3.0$, 1.0). In contrast, taxa with $>50\%$ of the population relying on the Yellow Sea declined at an average of $-5.2\%$ year$^{-1}$ (95% CRI: $-6.5$, $-3.8$). This group includes the EAAF endemics great knot (*Calidris tenuirostris*) and far eastern curlew (*Numenius madagascariensis*; Table 1), both of which are listed as globally threatened taxa[27].

**Geographical patterns in non-breeding population trends**. We found no evidence that population trends varied markedly among non-breeding nodes in Australia and New Zealand, indicating that conditions in the non-breeding period did not account for population declines. Of the eight taxa that had sufficient data for node-level trend estimation, populations of six credibly decreased in abundance, with declines occurring in all four nodes (Fig. 3; Supplementary Table 2). Despite intervening distances of up to 5,000 km, population trends among nodes for each taxon differed by mean odds of $<6\%$ (Supplementary Table 3). None of the eight taxa credibly increased in abundance on any node (Fig. 3), suggesting that movement among nodes was unlikely to explain population declines. Decreasing habitat quality at non-breeding sites caused by urban expansion[28] and river diversion for agriculture[29] is unlikely to explain population trends because most resident coastal breeding shorebirds that co-occur with migrants have not declined[19].

**Discussion**
Our analyses indicate that reliance on the Yellow Sea is a leading driver of population declines in migratory shorebirds of the

**Table 1 | Flyway-level population trend estimates and 95% CRI for ten EAAF migratory shorebird taxa and their reliance on Yellow Sea tidal mudflats.**

| Taxon | Yellow Sea reliance | Trend (95% CRI) |
|---|---|---|
| *menzbieri* bar-tailed godwit* | 1.00 | **−0.061 (−0.087, −0.037)** |
| Far eastern curlew* | 0.95 | **−0.058 (−0.070, −0.046)** |
| Curlew sandpiper | 0.90 | **−0.075 (−0.095, −0.055)** |
| Great knot* | 0.90 | **−0.051 (−0.075, −0.025)** |
| Red knot | 0.90 | **−0.044 (−0.066, −0.022)** |
| Lesser sand plover | 0.70 | **−0.060 (−0.102, −0.020)** |
| *baueri* bar-tailed godwit* | 0.50 | **−0.014 (−0.022, −0.006)** |
| Terek sandpiper | 0.40 | −0.023 (−0.067, 0.020) |
| Red-necked stint* | 0.35 | −0.016 (−0.031, 0.000) |
| Grey-tailed tattler* | 0.03 | 0.011 (−0.021, 0.041) |

CRI, credible interval; EAAF, East Asian-Australasian Flyway. Yellow Sea reliance is the proportion of the EAAF population that stages in the Yellow Sea during northbound and southbound migration combined. Population trend estimates are posterior means of slope parameter $\beta$ from equation (4). An asterisk denotes taxa endemic to the EAAF. Boldface estimates indicate credibly declining taxa.

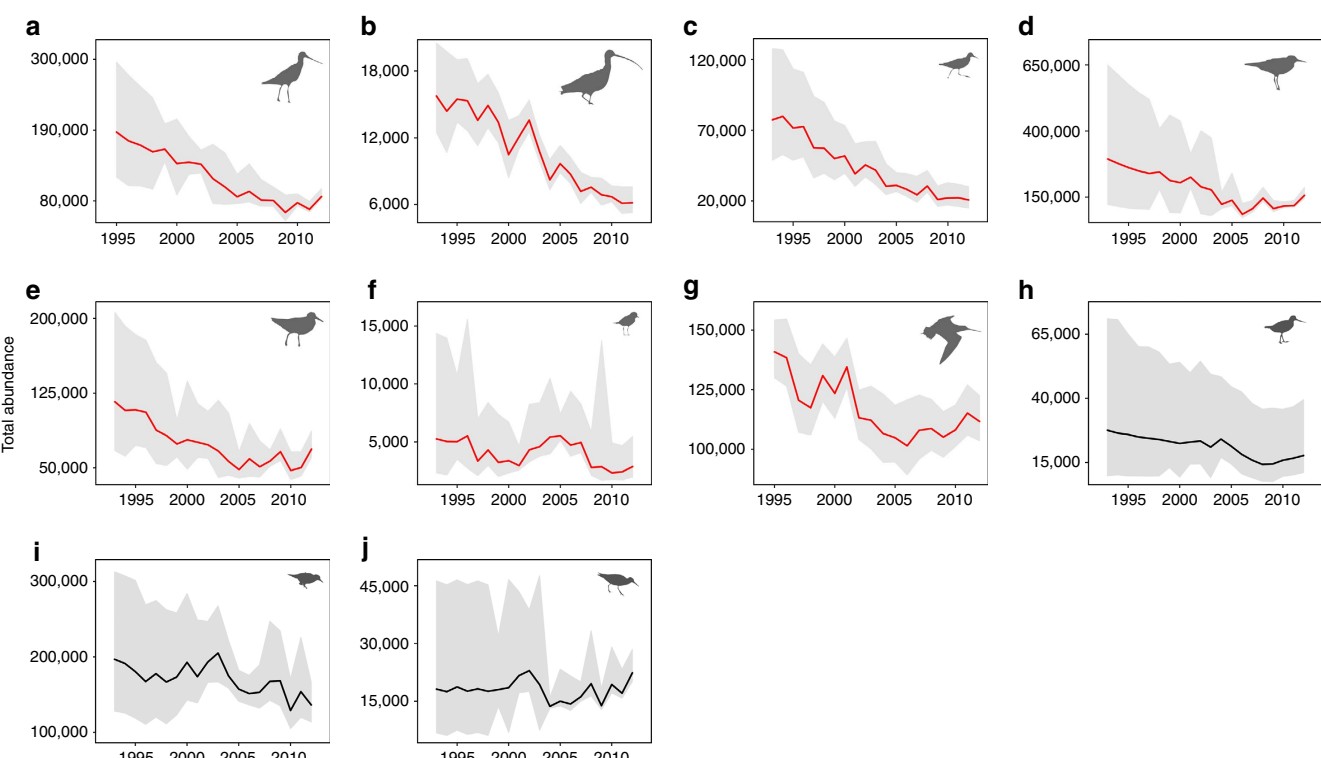

**Figure 2 | Total abundance between 1993 and 2012 for ten EAAF migratory shorebird taxa.** (**a**–**f**) Taxa are ordered from highest to lowest Yellow Sea reliance, the proportion of the flyway population that stages on Yellow Sea tidal mudflats to refuel for long-distance migrations. (**a**) *Menzbieri* bar-tailed godwit; (**b**) far eastern curlew; (**c**) curlew sandpiper; (**d**) great knot; (**e**) red knot; (**f**) lesser sand plover; (**g**) *baueri* bar-tailed godwit; (**h**) terek sandpiper; (**i**) red-necked stint; and (**j**) grey-tailed tattler. Total abundance estimates are posterior means from Bayesian N-mixture models of counts across Australia and New Zealand, including the majority of internationally important sites. Lines show posterior mean abundance estimates for each year, with red lines indicating taxa with credibly declining populations and grey shading denoting the 95% CRI. Overall trend estimates appear in Table 1. Detection probabilities for each taxon ranged from 0.52 to 0.68 (Supplementary Fig. 1) and were reflected in modelled abundances and trend estimates. Posterior predictive checks indicated good model fit in all cases (Supplementary Fig. 2).

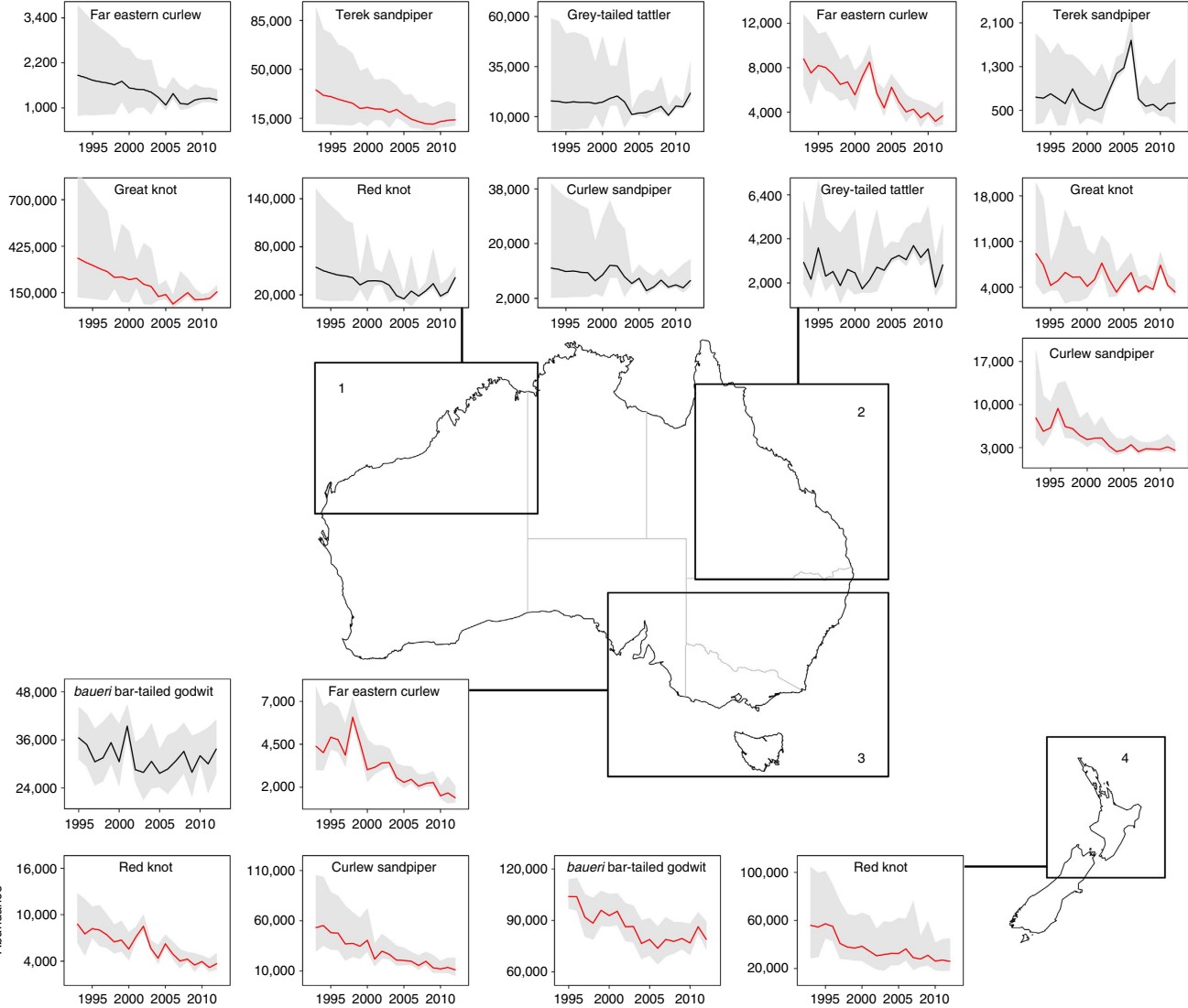

**Figure 3 | Abundance between 1993 and 2012 for eight EAAF migratory shorebird taxa found in four non-breeding population nodes.** Abundance estimates are posterior means for each node from Bayesian N-mixture models of counts across Australia and New Zealand, including the majority of internationally important sites. Lines show posterior mean abundance estimates for each year, with red lines indicating credibly declining populations and grey shading denoting the 95% CRI. Map insets indicate the location of the four population nodes: (1) northwestern Australia; (2) Queensland; (3) southeastern Australia and (4) New Zealand. Lesser sand plover and the *menzbieri* subspecies of bar-tailed godwit were not included in analyses because they occurred on only one node.

EAAF. This finding is consistent with previous work showing that survival estimates across the annual cycle for three taxa that underwent some of the largest population declines are dramatically lower during the Yellow Sea staging period compared with the stationary breeding and non-breeding periods[30]. This result implies that population declines are driven by low survival during or soon after staging in Yellow Sea tidal mudflats, likely because birds are unable to refuel enough to meet the energetic demands of migration. Sustained reproductive failure is unlikely to explain declines because the ratio of juvenile to adult shorebirds, an indicator of breeding success, has remained relatively constant at non-breeding sites in northwestern and southeastern Australia in the six taxa for which data are available[31,32]. Previous research in Japan has also shown that shorebird species relying on the Yellow Sea while on migration are declining more quickly than those that do not[11]. However, Japan itself is a stopover site for some species and is on the periphery of the migratory connectivity networks of the taxa we found to be declining fastest, complicating

interpretation of those earlier results. We have built substantially on this work by showing that Yellow Sea reliance explains variation in population trends at non-breeding destinations that support the majority of the flyway populations of ten taxa.

Two sources of long-term environmental decline in the Yellow Sea are likely responsible for the negative impacts on migratory shorebird populations. First, nearly 30% of Yellow Sea tidal mudflats were lost to coastal development in the past 30 years[33], a period that brackets the timing of observed population declines. Coastal China is forecast to undergo up to 14% expansion in urban development over the next 15 years, much of it concentrated on the margins of the Yellow Sea[34], and tidal mudflat loss seems likely to continue or accelerate. Second, the Yellow Sea ecosystem has undergone pervasive degradation in quality[35], including massive algal blooms[36], discharge of heavy metals and pesticides[37]; and the spread of the exotic saltmarsh-grass *Spartina alterniflora*[16]. These disturbances reduce prey availability and foraging opportunities for migrating shorebirds[38].

An enormous conservation intervention is needed to reverse the declines in EAAF shorebirds, and indeed an international governance framework is emerging in the region. A series of bilateral migratory bird agreements complements the EAAF Partnership (EAAFP), a collaboration among 34 countries, non-governmental organizations and multilateral arrangements. The EAAFP has thus far formally designated 136 Flyway Network Sites in response to the need for international coordination of habitat protection. Our results suggest that the most immediate priority remains preventing further tidal mudflat loss in the Yellow Sea region. Other management options include restoring lost habitat[39] and modifying existing artificial coastal habitats such as salt ponds, rice paddies, wastewater treatment plants and seawalls to maximize shorebird foraging and roosting opportunities. For example, traditional salt-pans can support substantial numbers of non-breeding shorebirds[40]. Although artificial habitats may not provide the most specialized taxa with an adequate substitute for intact tidal mudflats, options will be increasingly limited if Yellow Sea habitat loss continues. Our findings indicate an urgent need for effective conservation actions in this flyway to prevent further population declines in the migratory shorebirds it supports.

## Methods

**Study species.** We selected ten shorebird taxa for which detailed information on migratory connectivity was available for the EAAF. Experts with extensive flyway experience developed migratory connectivity networks that defined movements among internationally important wetland sites. Drawing on population surveys, mark-resight efforts, and daylight-level geolocator and satellite tracks from across the flyway, experts grouped individual sites into the smallest possible set for which birds were thought to follow unique migration routes through the EAAF. These relationships were depicted as a series of nodes that defined each group of sites connected by a set of edges that traced migration routes (see supplementary material in (ref. 22)). Experts reached consensus on network structure for ten taxa: bar-tailed godwit (two subspecies here analysed separately: *Limosa lapponica baueri* and *Limosa lapponica menzbieri*), far eastern curlew (*Numenius madagascariensis*), Terek sandpiper, (*Xenus cinereus*), grey-tailed tattler (*Tringa brevipes*), great knot (*Calidris tenuirostris*), red knot (two subspecies here analysed together: *Calidris canutus piersmai* and *Calidris canutus rogersi*), curlew sandpiper (*Calidris ferruginea*) and lesser sand plover (*Charadrius mongolus*). A connectivity map was unavailable for red-necked stint (*Calidris ruficollis*), but we included this species in analyses because it is one of the most abundant migratory shorebirds in Australia during the non-breeding period.

**Data acquisition.** We acquired count data for each taxon from local and national databases in Australia and New Zealand maintained by the Australasian Wader Studies Group, BirdLife Australia, BirdLife Western Australia, BirdLife Tasmania, Bird Observer's Club of Australia, Friends of Shorebirds SE, Friends of Streaky Bay District Parks, Hunter Bird Observers Club, New South Wales Wader Study Group, Ornithological Society of New Zealand, Queensland Wader Study Group, South Australian Ornithological Association and Victorian Wader Study Group. Members of these groups conducted counts at coastal sites during the non-breeding period from October–March while the shorebirds are in the Southern Hemisphere. Movements among non-breeding sites are likely minimal during this period, so counts in these months should represent the number of birds available for detection. Each site consisted of one or more discrete roosts where shorebirds reliably congregate during high-tide cycles that inundate tidal mudflats used for foraging (Supplementary Table 2). Experienced observers timed visits to coincide with high tide and counted shorebirds at all roosts within a site. Any local movement among roosts therefore minimally affect counts.

Shorebirds were counted at the majority of sites at least twice each Austral summer. We focused our analyses on counts in January and February, but supplemented them with counts from November, December and March if necessary. Shorebirds arrive earlier from southbound migration at sites in northwestern Australia compared to other sites in Australia[41], so counts there began in October–December to avoid counting during wet season rains that could have lowered detection probability[42]. For each taxon, we included sites with data likely to yield sufficient power to detect temporal trends in abundance, recognizing that the degree of aggregation varies among species. For example, red-necked stint is rather broadly distributed, while great knot congregates strongly, and inclusion of different sites or time series could alter resulting population trend estimates and CRIs. We included sites with overall maxima of at least 500 individuals of each taxon because sites with lower maxima had numerous years when no birds were recorded. At many sites, counts began in the early 1980s, but counts at key sites in northwestern Australia, Queensland and New Zealand began in the early 1990s

(ref. 43). These criteria yielded 10–26 sites for each taxon across the non-breeding range in Australia and New Zealand with count data from 1993–2012 (Supplementary Table 2). Each site fell within a non-breeding node of at least one taxon. The location and size of nodes varied among taxon, but could be grouped into four areas: (1) northwestern Australia; (2) Queensland; (3) southeastern Australia and (4) New Zealand. For each taxon, one or two non-breeding nodes lacked count data owing to geographic isolation or absence of formal surveys. In all cases, these nodes were located in Southeast Asia and the Gulf of Carpentaria in Australia.

**Modelling abundance and population trends.** Our modelling goal was to estimate abundance and population trends of the 10 taxa using 20 years of count data across Australia and New Zealand. For each taxon, we first assessed population trends at each unique non-breeding node and then across all count sites to estimate flyway-level trends. Our model was an N-mixture model, formulated as a hierarchical Poisson regression that estimates parameters for the latent state of abundance by accounting for imperfect detection of individuals in spatially or temporally replicated counts[44]. Incorporating detection probability into abundance estimates is essential to remedy bias owing to the common failure to detect all individuals in counts (Supplementary Fig. 1)[45].

Shorebird counts occurred at $i$ $(1,\ldots,R)$ sites in $t$ $(1,\ldots,K)$ years with $j$ $(1,\ldots,N)$ monthly counts in each year. Counts $c_{ijt}$ can therefore be considered as a binomial random variable where the local abundance of birds available for sampling $N_{it}$ is observed with detection probability $P_{ijt}$:

$$c_{ijt} \sim \text{Bin}\left(N_{it}, P_{ijt}\right). \quad (1)$$

A Poisson distribution with mean $\lambda_{it}$, a standard distribution for modelling spatial or temporal variation in abundance from counts, described local abundance $N_{it}$:

$$N_{it} \sim \text{Poi}(\lambda_{it}). \quad (2)$$

We estimated parameters for the latent state of abundance $\lambda_{it}$ with a log-linear model. For node-based analyses, the expected log abundance was:

$$\log(\lambda_{it}) = \omega_i + \alpha_k + \beta_k(t - t^*) + \varepsilon_{it}, \quad (3)$$

where $\omega_i$ is a random effect for each count site, $\alpha_k$ is an intercept for each non-breeding node, $\beta_k$ is the linear trend at each node centred on mean year $t^*$ and $\varepsilon_{it}$ is a random effect accounting for extra-Poisson dispersion in abundance. We assigned standard vague prior distributions to all parameters determining $\lambda_{it}$. We assumed that $\omega_i$, $\alpha_k$, $\beta_k$ and $\varepsilon_{it}$ were normally distributed with mean 0 and precision $\tau$ (precision $= 1/\text{variance}$) and that $\tau$ was inverse gamma distributed with shape and scale $10^{-3}$. To estimate population trends across all count sites, we rephrased equation (3) without node-specific terms:

$$\log(\lambda_{it}) = \alpha_i + \beta(t - t^*) + \varepsilon_{it}. \quad (4)$$

The resulting slope $\beta$ is thus a single trend estimate for all sites counted across the non-breeding range in Australia and New Zealand, providing an estimate of the long-term flyway population trend for each taxon.

We modelled detection probability $P_{ijt}$ as:

$$\text{logit}(P_{ijt}) = \gamma_t + \delta_{ijt}, \quad (5)$$

where $\gamma_t$ is the mean detection probability in year $t$ and $\delta_{ijt}$ is a random effect allowing detection probability to vary at site $i$ in month $j$ and year $t$. We assumed that $\gamma_t$ was normally distributed with mean 0 and variance $10^3$ and that $\delta_{ijt}$ was uniformly distributed with range $0 - 100$. We did not model predictors of detection probability by fitting covariates because data on weather, observer identity and similar variables were unavailable for most counts over the 20-year time series.

We fit N-mixture models in program JAGS[46] through the R2jags package[47] in program R (ref. 48). We ran three chains of each model for 800,000 iterations and discarded the initial 200,000 as burnin. We thinned each chain by 18, resulting in 100,000 posterior samples. Convergence was assessed with Gelman–Rubin diagnostics and was unambiguous in all cases. We judged population trends as credibly increasing or decreasing if their 95% CRIs did not overlap zero. We evaluated variation in population trends among nodes individually for each taxon. During model fitting, we derived pairwise odds ratios and 95% CRI and used them to assess differences in trends among nodes.

There are four key assumptions accompanying this model[45]. First, populations within each site are closed to immigration and emigration between monthly counts. Within-year movement of individually marked birds among a subset of monitored count sites was exceedingly low, suggesting that our data meet this assumption[49]. The second assumption is that detection probability is constant for all individuals within each taxon. Although we cannot assess this assumption directly, birds in each taxon likely were detected with equal probability because counts took place when individuals congregated at multispecies high-tide roosts. Third, abundance and detection probability are modelled with appropriate parametric distributions. We assessed this assumption through posterior predictive checks. This approach compares discrepancies in the observed data to discrepancies in replicate data simulated from the posterior distribution to diagnose model fit. Our measure of discrepancy was a chi-square goodness of fit test, a standard statistic used for posterior predictive checks[45]. The proportion of Markov Chain Monte Carlo (MCMC) samples where discrepancies in the observed data exceed those of the replicate data should be near 0.50 when model fit is adequate (Supplementary Fig. 2).

The fourth assumption is that double counts of individuals are minimal. Violation of this assumption would overestimate abundance and bias estimates of log-linear decline rates towards zero. All counts were conducted by experienced observers, who designed surveys to minimize overlooking or double-counting shorebirds, so our data likely meet this assumption.

**Predicting variation in flyway-level population trends.** We performed a phylogenetic comparative analysis to evaluate the ability of five different taxon-level predictors to explain the variation in flyway-level population trend estimates: Yellow Sea reliance, migration distance, breeding range size, generation time and body size. This analysis accounted for the possibility that shared evolutionary history partly could explain population declines. To develop our index of Yellow Sea reliance, we tallied the proportion of the flyway population for each taxon that stages[14] on tidal mudflats within the Yellow Sea region ($34°29'$ N, $119°47'$ E to $35°20'$ N, $129°17'$) based on published migratory connectivity networks[22] and estimates of shorebird populations staging in the Yellow Sea[23,24]. Because tidal mudflats in this region are declining at $>1$ % annually[16,33], we predicted that taxa with higher reliance on Yellow Sea staging sites would decline at a faster rate compared to those with lower reliance on this area. Ideally, this analysis would involve time series estimates of tidal mudflat loss that are contemporaneous with shorebird counts. However, the best available estimates of tidal mudflat loss are derived from only two time points[33,50], and annual estimates of habitat loss cannot be determined. To estimate differences in migration distance among taxa, we used digitized distribution maps[51] and a geographic information system[52] to measure the degrees of latitude travelled during migration as the latitudinal difference between the centroid of breeding and non-breeding ranges. We calculated breeding range size by digitizing published distribution maps[51] and acquired data on generation time and body size (measured from bill to tail) from regional handbooks[53,54] and published reports[55]. Species with long migrations, small breeding ranges, long generation times and large body size may face a higher risk of population decline or extinction[25,26]. We therefore predicted that taxa with geographical and life history attributes closer to these extremes would decline at a higher rate compared to those further away from these extremes.

To incorporate uncertainty into the phylogenetic comparative analysis, we acquired 1,000 equally plausible, ultrametric trees based on the Ericson backbone[56] from the BirdTree website[57]. Sequence data were not available for bar-tailed godwit subspecies, so trees contained a single branch for this species. We therefore performed two versions of the phylogenetic comparative analysis, one using predictors of population trends for *L. l. menzbieri* and another with predictors for *L. l. baueri*. These analyses yielded qualitatively similar results. We retained the analysis using *L. l. menzbieri*, the subspecies that showed the greater population decline. Simulations investigating tree samples size indicate that analyses of $>1,000$ trees does not yield more precise parameter estimates[58]. We used the ape package[59] in program R (ref. 48) to convert each tree to a variance–covariance matrix under the assumption of a Brownian Motion model of character evolution. Under this assumption, the variance is the branch length from the root to tip and the covariance is the branch length from the root to the most recent common ancestor of each pair of taxa[60]

The phylogenetic comparative analysis was a Bayesian hierarchical linear regression that evaluated which predictors best explained the differences in flyway population trends among $z$ $(1,\ldots, M)$ taxa and $q$ $(1,\ldots, R)$ trees. The flyway population trend for each taxon $Td_z$ estimated from equation (4) was drawn from a multivariate normal distribution with mean $\mu_z$ and variance $\sigma^2\Omega$:

$$Td_z \sim \mathrm{Mnorm}\left(\mu_z, \sigma^2\Omega\right) \quad (6)$$

where $\Omega$ is an array of dimension $M \times M \times R$ in which each element $\Omega\,[\,,\,,q]$ is the inverse of the scaled variance–covariance matrix for each tree $q$. We assumed that the variance $\sigma^2$ was inverse gamma distributed with shape and scale $10^{-3}$. With $\Omega$ scaled to a branch height of 1, $\sigma^2$ can be considered as residual variance and $\Omega$ as a correlation structure. Thus, larger values of $\Omega$ would to limit the ability of predictors to explain variation in population trends $Td_z$. The model for predicting flyway population trend differences among taxa was:

$$\mu_z = \alpha + \sum_{c=1}^{C} \beta_c X_{zc} + \varepsilon, \quad (7)$$

where $\alpha$ is an intercept, $\beta_c$ is the effect of covariate $c$, $X_{zc}$ is a vector of covariates for Yellow Sea reliance and predictors of population trends for taxon $z$, and $\varepsilon$ is an error term to account for unexplained variability. We used Bayesian variable selection to assess which predictors were most important for explaining the differences in flyway population trends among taxa. Bayesian variable selection is one approach for choosing important predictors from a candidate set and for developing parsimonious models[61,62]. This method is also useful for identifying influential covariates when wide 95% CRI indicate high estimation uncertainty[62]. To perform Bayesian variable selection, we defined $\beta_c$ as the product of a binary indicator variable $\upsilon_c$ and a regression coefficient $\theta_c$ such that:

$$\beta_c = \upsilon_c\theta_c, \quad (8)$$

$$\upsilon_c \sim \mathrm{Bernouli}(0.5), \quad (9)$$

$$\theta_c \sim \mathrm{Norm}(\mu_c, \sigma_c). \quad (10)$$

The Bernoulli prior allows the indicator variable $\upsilon_c$ to take a value of 1 if covariate $c$ is an important predictor and a value of 0 otherwise during each MCMC iteration. A large posterior mean for a given $\upsilon_c$ would identify it as an important predictor, whereas a mean close to 0 would mark it as unimportant. We interpreted indicator posterior means $\geq 0.75$ as evidence that a covariate was an important predictor, those between 0.25 and 0.75 as inconclusive, and those $\leq 0.25$ as evidence that a covariate was unimportant[62]. We then repeated the analysis by fixing unimportant values of the indicator variable $\upsilon_c$ to 0, improving the ability of retained predictors to explain population trends[62]. We assumed that $\alpha$, $\mu_c$ and $\varepsilon$ were normally distributed with mean 0 and variance $10^3$ and that $\sigma_c^2$ was inverse gamma distributed with shape and scale $10^{-3}$. We fit the regression model in program JAGS[46] through the R2JAGS package[47] in program R (ref. 48). We ran three chains of the model for 120,000 iterations, discarding the initial 20,000 as burnin. We thinned each chain by three, yielding 100,000 posterior samples.

**Code availability.** Program JAGS code for Bayesian models that support the findings of this study is available from https://github.com/studdsc/eaaf.shorebirds.

**Data availability.** The data that support the findings of this study are available from the organizations listed in the Acknowledgements and in Supplementary Note 1, but restrictions apply to the availability of these data, which were used under agreement for the current study. The authors will work upon request with any interested parties to try to secure data sharing agreements with the aforementioned organizations.

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

## Acknowledgements

This work is only possible because of the efforts of hundreds of people who have counted shorebirds across Australia and New Zealand, and those who have collated and curated those data; we list these in Supplementary Note 1 with grateful thanks to additional people who do not appear on this list. For data contributions we are grateful to: Australasian Wader Studies Group, Bird Observers Club of Australia, BirdLife Australia and their Shorebirds 2020 team, BirdLife Western Australia, BirdLife South Australia, BirdLife Tasmania, Hunter Bird Observers Club, Monitoring Yellow Sea Migrants in Australia, Ornithological Society of New Zealand, Queensland Wader Study Group, Victorian Wader Study Group and Wetlands International. Financial support was provided by Australian Research Council Linkage grant LP100200418, a Future Fellowship to R.A.F., a Federation Fellowship to H.P.P., the Queensland Wader Study Group, the Department of Environment Heritage and Protection (Queensland), the Department of the Environment and the Port of Brisbane. C.J.H. was supported by the major funders of the Global Flyway Network, including BirdLife Netherlands (2007–2012), WWF Netherlands (2010–2014, 2016) and the Spinoza Premium of Netherlands Organization Prize for Scientific Research to Theunis Piersma (2014–2016).

## Author contributions

C.E.S. and R.A.F. wrote the manuscript, with contribution from all authors. C.E.S. and N.J.M. conducted the statistical analyses. R.S.C., K.G., C.J.H., D.S.M., D.A.M., C.D.T.M., A.C.R., D.I.R., E.J.W. and R.J. contributed to data collation and interpretation. R.A.F., B.E.K., H.B.W. and H.P.P. conceived of and funded the research.

## Additional information

**Competing interests:** The authors declare no competing financial interests.

