## [Peer review file · Nature Communications]

Reviewers' comments:

Reviewer #1 (Remarks to the Author):

In this important piece of work, the authors investigate interspecific variation in population trends among shorebirds migrating through the east Asian and Australasian flyway. Based on long-term citizen science count data, they convincingly demonstrate that those species that make the largest use of Yellow Sea tidal mudflats as stopover sites during their migratory journeys have shown the strongest population declines, while those species that use such tidal flats to a lesser extent have declined the least. Given the increasing predicted human pressure on such habitats in the near future, results from this study suggests that, in the absence of immediate and concrete conservation actions, some species of migratory shorebirds may rapidly be driven to near extinction because of habitat deterioration. From an ecological viewpoint, this paper also suggests that the extent and quality of stopover habitat can be a major driver of population dynamics among migratory birds. Importantly, by comparing trends between the different wintering populations, the authors could rule out the effects of habitat changes in the wintering grounds in affecting population dynamics.

I think the manuscript is generally well written, although I advise to improve clarity of writings in several points (I will provide some suggestions below). I also have some remarks on the presentation and one major comment on the statistical analyses. On the whole, I think my comments should be rather easy to address in a revised version. I will list my detailed comments in order of appearance in the manuscript and mark major comments with M.

Title: I personally do not like the emphatic title the authors have proposed, as I think it is not sufficiently descriptive and accurate. How could a 'migratory flyway' actually 'collapse'? The flyway in itself cannot collapse! It is not even clear that the manuscript is dealing with migratory birds! I strongly advise modifying it, and suggest the following: Dramatic population decline in migratory shorebirds relying on Yellow Sea tidal mudflats as stopover sites.

Abstract: I think it is rather poorly written and should be almost entirely restructured. Please note that every single word in abstract for such high impact journals should be carefully evaluated and is of fundamental importance. As such, this abstract does not provide enough information to exactly understand what was made for the non specialist reader. For instance, the first sentence is totally useless to introduce this article. The term 'natural experiment' is misleading and unclear. There is no need at all to mention this.

Consider the following (to be adjusted to word limits):

Migratory animals are becoming increasingly threatened by human-induced global changes. However, little... Based on continent-wide citizen science data, we assess population trends of shorebirds migrating through the Yellow Sea tidal mudflats, an increasingly threatened ecosystem that has declined in extent by >65% in recent decades. We discovered that shorebird taxa...20 years compared to other species. This is....that stopover habitat... can considerably limit populations of multiple species of migratory birds....

Introduction

L49: Migratory animals undertake...demanding journeys

L53-54: This is not true. There are many studies of Afro-Palearctic migrants showing that stopover habitat quality (in terms of e.g. primary productivity, as gauged from rainfall or NDVI) can limit population dynamics. For instance, many studies link population dynamics with rainfall or NDVI in North Africa or in the Sahel stopover areas. These studies ought to be cited here.

L60: complete the tight annual schedules

L61: is increasingly being degraded

L70: Yellow Sea reliance is unclear to me, as well as the term 'natural experiment'. I would modify the sentence (e.g. variation in the use of the Yellow Sea tidal mudflats ...provides a unique opportunity to investigate whetherhabitat drives population....)

L89-93: This definitely belongs to the Methods section!

Results

L96: was the single most important...

L97: 'indicator value' is not clear and should be removed from here. It should be added to a figure legend with adequate explanation and reference to the Methods section.

L98: CRI should be spelled out in its first occurrence.

L101: what are the values reported in brackets? Please specify

L104: which are listed as ...threatened taxa.

L113: the red-necked

L156-157: I think this sentence goes beyond what a scientific article should say. I would rephrase it (e.g. Our study calls for urgent and effective concerted actions to improved the conservation status of this...)

Methods

L159: taxa that had

L202: began in the early

L206: sorted?

L213: We relied on N-mixture...as a hierarchical...

M L290: I think that any analysis comparing different species should account for shared

evolutionary history. You should include phylogenetic history in your analyses (there are several possibilities you can choose that best suit to your Bayesian modeling framework; please check out the recent book by L. Garamszegi on modern phylogenetic methods, Springer-Verlag). You might use BirdTree to get phylogenetic information for your analyses. I am quite confident this would not affect your findings, but will for sure strengthen your analyses.

P24: I think that Figure S3 is much more informative than Table 2 and should be included in the main text instead of Table 2. I would rather report the trends as insets within each small figure. The table can be reported in the supplementary material instead.

Reviewer #2 (Remarks to the Author):

This is an important and interesting paper dealing with what is clearly becoming a major conservation issue. It shows how a range of shorebirds have declined and that the rate of decline is linked to the use of the Yellow Sea, an area that is subject to a remarkable degree of habitat loss.

The methods are appropriate and it is done well.

40 The claim that 'This is the first demonstration that migratory stopover sites can limit populations of multiple species on a flyway' is clearly not true as this was shown by Amano et al (2010) who in the abstract state: 'dependence on the Yellow Sea negatively affected the population trends of waders'. My suggestion is that this is acknowledged and stated how it builds on that study.

Amano, T., Székely, T., Koyama, K., Amano, H. & Sutherland, W.J. 2010. A framework for monitoring the status of populations: An example from wader populations in the East Asian - Australasian flyway. *Biological Conservation* 143: 2238-2247.

68 minor point - I would make explicit that this is a subspecies

100-102 this seems a curious way of describing what looks much like a linear relationship.

104-106. I would make it clear what this means. With current loss will presumably reach a new equilibrium well above the described levels. This will need continued loss or deterioration of habitat.

126-9. This is a really powerful argument. Is there any way to show the data on Figure 2?

284 I have looked at the handbook and it is difficult to see how the estimate of generation time was generated from the limited data - unless there is much more in reference 47. It needs to describe how 'body size' was quantified such as the measure used and assume mean across the sexes.

153-157. This seems the wrong tone. It talks of extinctions and terminal collapse. This seems unlikely in the near future for the species analysed.

Furthermore this section makes a call to action, which I agree with but suggest is inappropriate here. I think the paper should concentrate on the science. They can make the call for action elsewhere. It could be reasonable to speculate as what might happen without action now.

I would expect this paper to play an important role in the conservation of this area and attract some publicity.

Bill Sutherland (signed as an author of Amano et al)

Reviewer #3 (Remarks to the Author):

Overall, I found this to be an extremely well written paper, with results of global significance for conservation of migratory species. The abstract is clear and succinct and summarizes key results.

As far as I can tell, the statistics are sound and appropriate for the available data sets. The N-mixture models used in the analysis are probably the most robust statistical approaches currently available for dealing with these types of count data (although I suspect that more or less the same conclusions would have been reached treating the counts as simple indices).

Obviously, as in any large scale ecological study, the results are somewhat correlative - it is impossible to get replicates at a hemispheric scale study.

However, the analyses fairly robustly examine alternative hypotheses, especially related to wintering areas, as well as one aspect of the breeding area (extent of breeding range).

One small improvement would be to discuss a bit more the elimination of hypotheses related to changes in the breeding grounds. Currently, this is not emphasized in the introduction, although, as noted, one feature is included in the model (extent of breeding range) and one more is mentioned in the discussion (lack of change in age ratios, albeit based on data from a 10-year old publication - not clear if that included data for all species in the current study).

Although I think the paper is quite sound as-is, the robustness of the paper would be strengthened even further if they could include a brief discussion of whether species with different reliance on the Yellow Sea have differences in breeding ranges, and whether they differ in their age ratios (although it is worth noting that differences in age-ratios could still be influenced by staging areas, as the breeding success of birds could be influenced by energetic condition upon arriving at the breeding grounds).

I strongly suspect that this additional discussion would serve only to strengthen the conclusions, as the evidence that staging area has a strong impact on survival is very strong.

Reviewers' comments:

Reviewer #1 (Remarks to the Author):

In this important piece of work, the authors investigate interspecific variation in population trends among shorebirds migrating through the East Asian and Australasian flyway. Based on long-term citizen science count data, they convincingly demonstrate that those species that make the largest use of Yellow Sea tidal mudflats as stopover sites during their migratory journeys have shown the strongest population declines, while those species that use such tidal flats to a lesser extent have declined the least. Given the increasing predicted human pressure on such habitats in the near future, results from this study suggests that, in the absence of immediate and concrete conservation actions, some species of migratory shorebirds may rapidly be driven to near extinction because of habitat deterioration. From an ecological viewpoint, this paper also suggests that the extent and quality of stopover habitat can be a major driver of population dynamics among migratory birds. Importantly, by comparing trends between the different wintering populations, the authors could rule out the effects of habitat changes in the wintering grounds in affecting population dynamics.

I think the manuscript is generally well written, although I advise to improve clarity of writings in several points (I will provide some suggestions below). I also have some remarks on the presentation and one major comment on the statistical analyses. On the whole, I think my comments should be rather easy to address in a revised version. I will list my detailed comments in order of appearance in the manuscript and mark major comments with M.

We thank the reviewer for this positive assessment.

Title: I personally do not like the emphatic title the authors have proposed, as I think it is not sufficiently descriptive and accurate. How could a 'migratory flyway' actually 'collapse'? The flyway in itself cannot collapse! It is not even clear that the manuscript is dealing with migratory birds! I strongly advise modifying it, and suggest the following: **Dramatic population decline in migratory shorebirds relying on Yellow Sea tidal mudflats as stopover sites.**

We have changed the title of the manuscript to “Dramatic population decline in migratory shorebirds relying on Yellow Sea tidal mudflats as stopover sites” as suggested by the reviewer.

Abstract: I think it is rather poorly written and should be almost entirely restructured. Please note that every single word in abstract for such high impact journals should be carefully evaluated and is of fundamental importance. As such, this abstract does not provide enough information to exactly understand what was made for the non-specialist reader. For instance, the first sentence is totally useless to introduce this article. The term 'natural experiment' is misleading and unclear. There is no need at all to mention this.

Consider the following (to be adjusted to word limits):

Migratory animals are becoming increasingly threatened by human-induced global changes. However, little... Based on continent-wide citizen science data, we assess population trends of shorebirds migrating through the Yellow Sea tidal mudflats, an increasingly threatened ecosystem that has declined in extent by >65% in recent decades. We discovered that shorebird taxa...20 years compared to other species. This is....that stopover habitat... can considerably limit populations of multiple species of migratory birds....

We have thoroughly rewritten the Abstract, incorporating several of the specific recommendations.

Introduction

L49: Migratory animals undertake...demanding journeys

Changed as suggested (line 50).

L53-54: This is not true. There are many studies of Afro-Palaearctic migrants showing that stopover habitat quality (in terms of e.g. primary productivity, as gauged from rainfall or NDVI) can limit population dynamics. For instance, many studies link population dynamics with rainfall or NDVI in North Africa or in the Sahel stopover areas. These studies ought to be cited here.

We now cite two studies that address how primary productivity at stopover sites can influence population dynamics of Palaearctic migrants (refs 12 & 13).

L60: complete the tight annual schedules

Changed as suggested (line 61).

L61: is increasingly being degraded

Changed as suggested (line 62).

L70: Yellow Sea reliance is unclear to me, as well as the term 'natural experiment'. I would modify the sentence (e.g. variation in the use of the Yellow Sea tidal mudflats ...provides a unique opportunity to investigate whetherhabitat drives population....)

We have carefully rewritten this sentence (lines 70-72).

L89-93: This definitely belongs to the Methods section!

We agree that this information is not suitable for the Introduction. However, we believe that Supplementary Data Figs. 1 & 2 give important context for the results, given that estimates of detection probability directly inform modeled abundance and that the posterior predictive tests indicate good model fit. We have therefore moved these sentences to the legend in Figure 2. We also include reference to these Figures in the Methods as suggested (lines 219 & 272) .

Results

L96: was the single most important...

Changed as suggested (line 95).

L97: 'indicator value' is not clear and should be removed from here. It should be added to a figure legend with adequate explanation and reference to the Methods section.

We have removed the mention of indicator values from the Results and now treat the indicator value results in the Figure 1 legend. We have also worked to clarify the explanation of this analysis in the Methods (lines 316-334).

L98: CRI should be spelled out in its first occurrence.

Changed as suggested (line 97).

L101: what are the values reported in brackets? Please specify

The values in brackets are 95% credible intervals (CRI). We have revised the text to note this (lines 100-102).

L104: which are listed as ...threatened taxa.

Changed as suggested (line 103).

L113: the red-necked

When double-checking the results, we discovered that we had erroneously written the mean and 95% CRI for the red-necked stint for node 3 as 0.018 (0.009, 0.045), instead of 0.018 (-0.010, 0.045). So, in fact none of the taxa credibly increased in abundance in any node, and we have updated the text accordingly (lines 111-112).

L156-157: I think this sentence goes beyond what a scientific article should say. I would rephrase it (e.g. Our study calls for urgent and effective concerted actions to improved the conservation status of this...)

We have amended the closing sentence to restrict it more closely to conclusions from the science (lines 157-159).

Methods

L159: taxa that had

Changed as suggested (line 161).

L202: began in the early

Changed as suggested (line 204).

L206: sorted?

We have changed “sorted” to “could be grouped” to reflect our role in the process of grouping nodes for the taxa (line 207).

L213: We relied on N-mixture...as a hierarchical...

We have rephrased the opening part of this sentence (line 215).

M L290: I think that any analysis comparing different species should account for shared evolutionary history. You should include phylogenetic history in your analyses (there are several possibilities you can choose that best suit to your Bayesian modeling framework; please check out the recent book by L. Garamszegi on modern phylogenetic methods, Springer-Verlag). You might use BirdTree to get phylogenetic information for your analyses. I am quite confident this would not affect your findings, but will for sure strengthen you analyses.

We have performed an analysis accounting for shared evolutionary history (lines 94-95 in results; lines 276 & 293-311 in Methods), and indeed while the parameter estimates change slightly, Yellow Sea Reliance remains the single most important predictor of the differences in population trends.

P24: I think that Figure S3 is much more informative than Table 2 and should be included in the main text instead of Table 2. I would rather report the trends as insets within each small figure. The table can be reported in the supplementary material instead.

As suggested, we have moved Supplementary Data Figure 3 to the main text as Figure 3 and have moved Table 2 to the Supplementary Information and Supplementary Data Table 3.

Reviewer #2 (Remarks to the Author):

This is an important and interesting paper dealing with what is clearly becoming a major conservation issue. It shows how a range of shorebirds have declined and that the rate of decline is linked to the use of the Yellow Sea, an area that is subject to a remarkable degree of habitat loss.

The methods are appropriate and it is done well.

40 The claim that 'This is the first demonstration that migratory stopover sites can limit populations of multiple species on a flyway' is clearly not true as this was shown by Amano et al

(2010) who in the abstract state: 'dependence on the Yellow Sea negatively affected the population trends of waders'. My suggestion is that this is acknowledged and stated how it builds on that study.

Amano, T., Székely, T., Koyama, K., Amano, H. & Sutherland, W.J. 2010. A framework for monitoring the status of populations: An example from wader populations in the East Asian-Australasian flyway. *Biological Conservation* 143: 2238-2247.

We have tempered the language in the abstract, and now correct our oversight of not citing Amano et al. (2010; lines 55, 66, 129). In addition, we state explicitly how we believe our work has built on Amano et al., and discuss this point further (lines 127-133).

68 minor point - I would make explicit that this is a subspecies

Changed as suggested (line 68).

100-102 this seems a curious way of describing what looks much like a linear relationship.

Although our analysis revealed a linear relationship between population trends and Yellow Sea reliance, we note that in Fig. 1B there is nonetheless an evident break in population trends between taxa with the very high Yellow Sea reliance and those with lower Yellow Sea reliance. While not the centerpiece of our results, we felt that including the observation that taxa with > 50% Yellow Sea reliance all have declining populations is an important detail that could provide context for future analyses.

104-106. I would make it clear what this means. With current loss will presumably reach a new equilibrium well above the described levels. This will need continued loss or deterioration of habitat.

We have removed these projections, considering them to be an unnecessary distraction in this paper.

126-9. This is a really powerful argument. Is there any way to show the data on Figure 2?

We would like to show age ratios of first year birds in the non-breeding period on a second y-axis in Figure 2, however we lack such data over a sufficient spatial scale to make a convincing presentation given that this figure shows continental scale population trends. Thus, we think the age ratio data are best treated in the discussion. This is clearly an area for further work.

284 I have looked at the handbook and is difficult to see how the estimate of generation time was generated from the limited data - unless there is much more in reference 47. It needs to describe how 'body size' was quantified such as the measure used and assume mean across the sexes.

Yes, there is much new information on generation time in references 49 (Garnett et al. 2010). Body size was measured from bill to tail as reported in reference 47 (Marchant & Higgins 1993) & 48 (Higgins & Davies 1996) and we have included this information in the Methods (line 288).

153-157. This seems the wrong tone. It talks of extinctions and terminal collapse. This seems unlikely in the near future for the species analysed.

Furthermore this section makes a call to action, which I agree with but suggest is inappropriate here. I think the paper should concentrate on the science. They can make the call for action elsewhere. It could be reasonable to speculate as what might happen without action now.

I would expect this paper to play an important role in the conservation of this area and attract some publicity.

We have amended the closing sentence to restrict it more closely to conclusions from the science (lines 157-159).

Bill Sutherland (signed as an author of Amano et al)

Reviewer #3 (Remarks to the Author):

Overall, I found this to be an extremely well written paper, with results of global significance for conservation of migratory species. The abstract is clear and succinct and summarizes key results.

As far as I can tell, the statistics are sound and appropriate for the available data sets. The N-mixture models used in the analysis are probably the most robust statistical approaches currently available for dealing with these types of count data (although I suspect that more or less the same conclusions would have been reached treating the counts as simple indices).

Obviously, as in any large-scale ecological study, the results are somewhat correlative - it is impossible to get replicates at a hemispheric scale study.

However, the analyses fairly robustly examine alternative hypotheses, especially related to wintering areas, as well as one aspect of the breeding area (extent of breeding range).

We thank the reviewer for this positive assessment.

One small improvement would be to discuss a bit more the elimination of hypotheses related to changes in the breeding grounds. Currently, this is not emphasized in the introduction, although, as noted, one feature is included in the model (extent of breeding range) and one more is mentioned in the discussion (lack of change in age ratios, albeit based on data from a 10-year old publication - not clear if that included data for all species in the current study).

As the reviewer notes, we have included a number of other variables as alternative possible predictors, yet space does not permit a full explanation of all of these. Consequently, we intended for the narrative of the introduction to focus down rather quickly on stopover sites as a key potential driver of decline in this flyway. However, in addition to the two discussion points noted by the reviewer (breeding range size and age ratios), we also discuss the survival estimates of Piersma et al (2016) suggesting that the major impacts on survival are located in stopover sites rather than on breeding or non-breeding grounds. We have clarified that the two papers we cite on age ratios cover data for 6 of the 10 taxa (line 127).

Although I think the paper is quite sound as-is, the robustness of the paper would be strengthened even further if they could include a brief discussion of whether species with different reliance on the Yellow Sea have differences in breeding ranges, and whether they differ in their age ratios (although it is worth noting that differences in age-ratios could still be influenced by staging areas, as the breeding success of birds could be influenced by energetic condition upon arriving at the breeding grounds).

I strongly suspect that this additional discussion would serve only to strengthen the conclusions, as the evidence that staging area has a strong impact on survival is very strong.

We have now included a brief discussion stating that there is no consistent difference in the location of breeding grounds for taxa with different reliance on the Yellow Sea (lines 84-87). Age ratio data from the breeding grounds are not available, and it is indeed interesting that there is no obvious change in age ratios in Australia given the fact that conditions on stopover sites can indeed contribute to breeding success, or survival of juveniles on their first migration.

Reviewers' comments:

Reviewer #1 (Remarks to the Author):

The authors have modified the manuscript according to the suggestions. Upon reading the ms again, I have however a few further comments that I would like the authors to address. I also make some specific suggestion for improving the clarity of the text.

L7: Yellow Sea as a stopover site showed

L9: among taxa

L10: I would rephrase: and was not significantly predicted by breeding range

L13: animal migrations are increasingly

L18: This is incorrect. There are many examples of very short distance migrations that are not forced to overcome ecological barriers. Specify that you are specifically referring to long distance migratory species here.

L32: species' declines

L41-42: unclear sentence. Consider: shorebird taxa for which expert-based migratory connectivity networks are available to test if...

L48: four nodes of

L52-53: this sentence is surprising. Needs to be tested quantitatively. Related to this, I have realized that you did not consider the migration distance as potential explanatory variable in your analyses of predictors of population decline. There are several studies showing that long-distance migratory birds are suffering stronger population declines than short-distance migrants. Although you have mostly long-distance migrants, some of the species you consider migrate further than others. Hence, I think that you should consider some indicators of migration distance as a further predictor in your model. Of course, this might be related to breeding latitude. Ideally, both breeding latitude and migration distance should be included as potential predictors, but I am not sure if you can consider both in your modelling exercise.

L56: consider rewording this paragraph: If Yellow Sea habitat changes drive population trends....with Yellow Sea reliance during migration, and taxon-specific....Alternatively, if environmental conditions in the non-breeding areas....changes in abundance should vary markedly [I think 'consistently' is confusing here]...reliance during migration.

I think that the precise reasons why you predict that coherent trends in the non-breeding grounds would support your hypothesis that Yellow Sea reliance is important in affecting population trends should be spelled out explicitly. You should include a further sentence to this last paragraph of the intro detailing why you think that variation in trends among nodes would support a stronger role for non-breeding nodes conditions in shaping population

trends. Is this because you assume that ecological conditions potentially affecting trends are not expected to covary among nodes? Please clarify.

L62: history among taxa, we found that

L62: Yellow Sea reliance needs to be defined here. Perhaps add in parenthesis something like (the estimated proportion of the EAAF population of a given taxon using Yellow Sea tidal mudflats as a stopover site).

L67: in table 1, YS reliance is expressed as a proportion, not as a percentage. Please use coherent notation (either proportions or percentages)

L69: with >50%

L73: varied markedly

L110: exotic coastal grass *Spartina*...[clarify what *Spartina* is]

L139-140: How did you consider *L. l. baueri* and *L. l. menzbieri* in your phylogeny? If you used BirdTree to get your phylogenetic trees, these two taxa are not separated. Please describe how you have dealt with this in L266 and subsequent.

L244: experienced observers

L258: you can easily calculate centroids of breeding and winter distributions from your maps to deal with previous comments.

L286: extinction risk? Do you mean population trends?

Fig. 1a: I am not familiar with Bayesian variable selection.

Fig. 1 b: there is no 'Bayesian linear regression' in this figure, just a simple scatterplot of population trends vs. YS reliance index. Otherwise, please add the regression line with CRI. Am I wrong?

Fig. 2 & 3: While looking more in detail at these figures, I was a bit confused. In the legend, you refer to 'flyway-level trend estimates', and I would expect to see the overall trend line for each taxon with a slope corresponding to the value reported in Table 1. Instead, I see broken lines depicting annual changes, which you name 'Trend lines': actually, this is not correct, as these broken lines are not 'trend lines'. How were these year-to-year changes and their CRI derived? This needs to be explained better. Moreover, according to your legends, these lines should show "% annual change in total abundance", but the y-axis does not show any % value.

Figure 3: you mention you did not show *menzbieri* and lesser sand plover. However, there is a graph for *menzbieri* (upper left corner), but not for lesser sand plover. Please check carefully for consistency and mistakes.

Reviewer #3 (Remarks to the Author):

Overall, I believe the authors have done an excellent job of addressing the reviewers comments, not only my own but, as far as I can tell, those of the other two reviewers as well.

I have one minor editorial comment:

One line 100 of the pdf, I think this should be worded “remained stable or declined only slightly”

A mean of -1% is still a decline, even if the CRI overlaps zero ... (and, of course, this is a mean across species suggesting some species may be declining slightly)

This doesn't change any other messaging.

I believe this is a very important paper, both in terms of understanding migration ecology, and also for supporting conservation.

Reviewers' comments:

Reviewer #1 (Remarks to the Author):

The authors have modified the manuscript according to the suggestions. Upon reading the ms again, I have however a few further comments that I would like the authors to address. I also make some specific suggestion for improving the clarity of the text.

L7: Yellow Sea as a stopover site showed

Changed as suggested (line 39).

L9: among taxa

Changed as suggested (line 42).

L10: I would rephrase: and was not significantly predicted by breeding range

We have rephrased to “not predicted by” (line 42).

L13: animal migrations are increasingly

Changed as suggested (line 45).

L18: This is incorrect. There are many examples of very short distance migrations that are not forced to overcome ecological barriers. Specify that you are specifically referring to long distance migratory species here.

Changed as suggested (line 46).

L32: species' declines

Changed as suggested (line 65).

L41-42: unclear sentence. Consider: shorebird taxa for which expert-based migratory connectivity networks are available to test if...

Changed as suggested (lines 75-76).

L48: four nodes of

Changed as suggested (line 82).

L52-53: this sentence is surprising. Needs to be tested quantitatively. Related to this, I have realized that you did not consider the migration distance as potential explanatory variable in your

analyses of predictors of population decline. There are several studies showing that long-distance migratory birds are suffering stronger population declines than short-distance migrants. Although you have mostly long-distance migrants, some of the species you consider migrate further than others. Hence, I think that you should consider some indicators of migration distance as a further predictor in your model. Of course, this might be related to breeding latitude. Ideally, both breeding latitude and migration distance should be included as potential predictors, but I am not sure if you can consider both in your modeling exercise.

We now include migration distance as a predictor in the analysis of the potential drivers of population trends, although it turned out not to be a strong predictor (lines 293-300, Figure 1A). As the reviewer suspects, breeding latitude and migration distance were indeed correlated, and analyzing both would have limited our power to detect effects. We agree that migration distance is the better choice of the two because it measures the length of migratory journey birds would travel to reach breeding areas of different latitude.

L56: consider rewording this paragraph: If Yellow Sea habitat changes drive population trends....with Yellow Sea reliance during migration, and taxon-specific....Alternatively, if environmental conditions in the non-breeding areas....changes in abundance should vary markedly [I think 'consistently' is confusing here]...reliance during migration.

Changed as suggested (line 92).

I think that the precise reasons why you predict that coherent trends in the non-breeding grounds would support your hypothesis that Yellow Sea reliance is important in affecting population trends should be spelled out explicitly. You should include a further sentence to this last paragraph of the intro detailing why you think that variation in trends among nodes would support a stronger role for non-breeding nodes conditions in shaping population trends. Is this because you assume that ecological conditions potentially affecting trends are not expected to covary among nodes? Please clarify.

We have clarified the prediction that taxon-specific changes in abundances should vary among non-breeding nodes if non-breeding conditions shape population trends by adding a sentence stating that declines should be confined to a subset of nodes where habitat quality is declining (lines 93-95).

L62: history among taxa, we found that

Changed as suggested (line 97).

L62: Yellow Sea reliance needs to be defined here. Perhaps add in parenthesis something like (the estimated proportion of the EAAF population of a given taxon using Yellow Sea tidal mudflats as a stopover site).

We are hesitant to disrupt the reporting of results with a parenthetical. We describe how we calculated Yellow Sea reliance in the preceding paragraph, eleven lines above (lines 83-85).

L67: in table 1, YS reliance is expressed as a proportion, not as a percentage. Please use coherent notation (either proportions or percentages).

We use proportions for all references to YS reliance, except the one noted in the text. We would prefer to keep this single use of percentages in discussing YS reliance because it allows us to be clear and concise in discussing the relationship between YS reliance and population trends. To make references to percentages more clear, we have changes the text to, “Taxa with <40% of the population relying on the Yellow Sea remained stable...” (line 102. We also have made this change to the text on line 104.

L69: with >50%

Changed as suggested (line 103).

L73: varied markedly

Changed as suggested (line 107).

L110: exotic coastal grass *Spartina*...[clarify what *Spartina* is]

We have changed this description to “exotic saltmarsh grass” (lines 144-145).

L139-140: How did you consider *L. l. baueri* and *L. l. menzbieri* in your phylogeny? If you used BirdTree to get your phylogenetic trees, these two taxa are not separated. Please describe how you have dealt with this in L266 and subsequent.

BirdTree indeed has sequence data only for bar-tailed godwit as a species and not separately for *L. l. menzbieri* and *L. l. baueri*, and it was an oversight not to include reference to this in the revision. We ran two versions of the analysis, one that used predictors of population trends for *L. l. menzbieri* and one that used predictors for *L. l. baueri*. The two analyses delivered very similar results. The report results for *L. l. menzbieri* because it gave a stronger test of the of the idea that shared evolutionary history could partly account for population declines given it experienced larger declines than *L. l. baueri* subspecies (lines 305-310).

L244: experienced observers

Changed as suggested (line 192).

L258: you can easily calculate centroids of breeding and winter distributions from your maps to deal with previous comments.

As described above, we have used centroids of breeding and non-breeding distribution to include migration distance in our analysis of the drivers of population trends (line 293-300).

L286: extinction risk? Do you mean population trends?

The reference used to support the selection of our predictors concerns understanding extinction risk. However, because we use these predictors to understand drivers of population trends, we have changed the text to “population trends” (line 331).

Fig. 1a: I am not familiar with Bayesian variable selection.

Bayesian variable selection is an approach for developing more parsimonious models by removing variables from consideration that have little explanatory power. It is motivated by the idea that some variables with 95% CRI close to or even overlapping 0 may nonetheless be important. We did not have such variables in the model, but we did not know this *a priori*. We describe how we use Bayesian variable selection in the analysis on lines 332-349 of the Methods.

Fig.1 b: there is no ‘Bayesian linear regression’ in this figure, just a simple scatterplot of population trends vs. YS reliance index. Otherwise, please add the regression line with CRI. Am I wrong?

We have added the regression line and 95% CRI in Figure 1B and have changed the figure legend accordingly.

Fig. 2 & 3: While looking more in detail at these figures, I was a bit confused. In the legend, you refer to ‘flyway-level trend estimates’, and I would expect to see the overall trend line for each taxon with a slope corresponding to the value reported in Table 1. Instead, I see broken lines depicting annual changes, which you name ‘Trend lines’: actually, this is not correct, as these broken lines are not ‘trend lines’. How were these year-to-year changes and their CRI derived? This needs to be explained better. Moreover, according to your legends, these lines should show “% annual change in total abundance”, but the y-axis does not show any % value.

It is understandable that the figure legends caused some confusion because they focused rather more on the interpretation of the figures than the figures themselves. Below, we first describe the analysis behind the figures and then summarize the changes we have made to increase the clarity of the figure legends.

The lines shown in Figures 2 & 3 are the posterior mean abundance estimates for each year and 95% CRI from equation 3 on line 235 (Figure 2) and from equation 4 on line 243 (Figure 3). As we describe on lines 232-245, we modeled abundance with extra poisson variation ϵ_{it} to account for the fact that abundance changes differently over time at different count sites. This term causes the posterior abundance values to track closely the observed abundance from counts, resulting in the broken lines and CRI.

The mean of the posterior means of slope β gives the mean annual rate of change in abundance, which is the population trend. This is equal to the posterior mean value of slope β (plus or minus poisson error) that we would get if we fit β as a straight line without the ϵ term. We have included a description of how trend estimates were calculated in the Table 1 notes (below the table).

With this in mind, we agree that the legend is incorrect in referring to population trends and we have therefore removed reference to population trends from both figure titles and legends in Figures 2 & 3. We now refer to the lines in these figures as, “Lines show posterior mean abundance estimates for each year”. We also have added to the Figure 2 legend the text “Overall trend estimates appear in Table 1.”

Figure 3: you mention you did not show menzbieri and lesser sand plover. However, there is a graph for menzbieri (upper left corner), but not for lesser sand plover. Please check carefully for consistency and mistakes.

We thank the reviewer for the bringing this error to our attention – we have now removed the *L. l. menziberi* plot.

Reviewer #3 (Remarks to the Author):

Overall, I believe the authors have done an excellent job of addressing the reviewers comments, not only my own but, as far as I can tell, those of the other two reviewers as well.

I have one minor editorial comment:

One line 100 of the pdf, I think this should be worded “remained stable or declined only slightly”
A mean of -1% is still a decline, even if the CRI overlaps zero ... (and, of course, this is a mean across species suggesting some species may be declining slightly)

We have changed the wording of this sentence as suggested (line 102).

This doesn't change any other messaging.

I believe this is a very important paper, both in terms of understanding migration ecology, and also for supporting conservation.

REVIEWERS' COMMENTS:

Reviewer #1 (Remarks to the Author):

I think the authors have addressed all the previous comments I raised.

I only have a few final remarks on the additions.

L306: You write: Sequence data were not available for bar-tailed godwit subspecies, so trees contained a single branch for this species. We therefore performed two versions of the phylogenetic comparative analysis, one using predictors of population trends for *L. l. menzbieri* and other with predictors for *L. l. baueri*. These analyses yielded similar results, but we report those using *L. l. menzbieri* because it experienced a greater change in abundance and thus provided a stronger test of the idea that shared evolutionary history partly could account for population declines.

I think this should be better as follows: Sequence data were not available for bar-tailed godwit subspecies, so trees contained a single branch for this species. We therefore performed two versions of the phylogenetic comparative analysis, one using predictors of population trends for *L. l. menzbieri* and another with predictors for *L. l. baueri*. These analyses yielded qualitatively similar results. We retained the analysis using *L. l. menzbieri*, the subspecies that showed the greatest population decline.

I am not sure the justification you suggest is correct. I think reporting the results using one or the other taxon should return qualitatively the same results.

Related to the above, please add in the legend to Fig. 1 that the analyses reported here refer to the analyses including *L. l. menzbieri* and excluding *L. l. baueri* (you might consider referring to the Methods to explain why).

REVIEWERS' COMMENTS:

Reviewer #1 (Remarks to the Author):

I think the authors have addressed all the previous comments I raised.

I only have a few final remarks on the additions.

L306: You write: Sequence data were not available for bar-tailed godwit subspecies, so trees contained a single branch for this species. We therefore performed two versions of the phylogenetic comparative analysis, one using predictors of population trends for *L. l. menzbieri* and other with predictors for *L. l. baueri*. These analyses yielded similar results, but we report those using *L. l. menzbieri* because it experienced a greater change in abundance and thus provided a stronger test of the idea that shared evolutionary history partly could account for population declines.

I think this should be better as follows: Sequence data were not available for bar-tailed godwit subspecies, so trees contained a single branch for this species. We therefore performed two versions of the phylogenetic comparative analysis, one using predictors of population trends for *L. l. menzbieri* and another with predictors for *L. l. baueri*. These analyses yielded qualitatively similar results. We retained the analysis using *L. l. menzbieri*, the subspecies that showed the greatest population decline.

We have changed the text as suggested.

I am not sure the justification you suggest is correct. I think reporting the results using one or the other taxon should return qualitatively the same results.

Related to the above, please add in the legend to Fig. 1 that the analyses reported here refer to the analyses including *L. l. menzbieri* and excluding *L. l. baueri* (you might consider referring to the Methods to explain why).

As suggested, we have changed the figure legend to note that the analyses include *L. l. menzbieri* and exclude *L. l. baueri*.